# Metabolic Syndrome and Colorectal Cancer Risk: Results of Propensity Score-Based Analyses in a Community-Based Cohort Study

**DOI:** 10.3390/ijerph17228687

**Published:** 2020-11-23

**Authors:** Jinsun Kim, Eun Young Park, Eunjung Park, Min Kyung Lim, Jin-Kyoung Oh, Byungmi Kim

**Affiliations:** 1Division of Cancer Prevention & Early Detection, National Cancer Control Institute, National Cancer Center, Goyang-si, Gyeonggi-do 10408, Korea; lilykim1011@gmail.com (J.K.); eunjungpark@ncc.re.kr (E.P.); jkoh@ncc.re.kr (J.-K.O.); kbm5369@ncc.re.kr (B.K.); 2Department of Cancer Control and Population Health, National Cancer Center Graduate School of Cancer Science and Policy, Goyang-si, Gyeonggi-do 10408, Korea; mickey@ncc.re.kr

**Keywords:** metabolic syndrome, colorectal cancer, propensity score methods, cohort

## Abstract

Background: This study aimed to determine the effects of metabolic syndrome (MetS) on colorectal cancer (CRC) using propensity score (PS) methods. Methods: The study subjects were 2417 men and 4568 women from the Korean National Cancer Center (KNCC) Community Cohort enrolled between 2003 and 2010. Odds risks (ORs) and 95% confidence intervals (CIs) using PS matching analysis, regression models adjusted by the PS or stratified into five strata according to PS, and PS weighting methods were calculated. Results: In women, MetS and abnormally high triglyceride (TG) levels were associated with CRC risk using the PS matching analysis (ORs, for MetS, 2.19 (95% CI, 1.10–4.33); for abnormal TG levels, 2.08 (95% CI, 1.07–4.02)). However, there were no significant associations between MetS and TG levels and CRC risk in men. Conclusions: Our study might provide additional evidence that deteriorated metabolic profiles increase the risk of CRC in women rather than men. Thus, this may have an important role in effective population-level interventions for deteriorated metabolic profiles at an early stage.

## 1. Introduction

Colorectal cancer (CRC) is the third most common cancer in both sexes worldwide (1,360,602 cases, 9.7% of the total cancer burden). According to GLOBOCAN 2018, the estimated age-standardized rates of CRC incidence for both sexes was observed to be similar patterns in most countries, ranking fourth in the United States, third in Europe, second in Japan, and third in the Republic of Korea [1].

Worldwide, over a billion people are known to be affected with metabolic syndrome (MetS). The MetS prevalence is increasing in low socio-economic countries as well as high socio-economic countries. In even young adults, the prevalence ranges from 5 to 7% worldwide, although it increases with age. MetS is a complex disorder characterized by a cluster of moderate levels of metabolic, anthropometric, and hemodynamic abnormalities, accepted as a modifiable risk factor CRC, although the mechanism linking MetS and CRC has not been clearly elucidted [2,3]. Recent systematic reviews and meta-analyses have conclusively reported that MetS is associated with an approximately 1.3-fold increased risk of CRC in both sexes, although the risk in women was slightly higher than that in men [2,4]. A few studies have been conducted in the Republic of Korea among national health insurance subscribers or subjects who underwent colonoscopy for a health examination in a hospital. However, these studies reported inconsistent results [5,6,7]. So far, there is limited evidence on the association between MetS and CRC, especially for the Asian population. Furthermore, the findings of previous observational studies have pointed inevitably lower causality than randomized controlled trials due to selection biases. In order to overcome this weakness of observational studies, propensity score (PS)-based methods were proposed to attenuate selection biases by balancing many covariates [8,9]. Studies using these methods have been increasingly published in a wide range of fields, including some observational studies on MetS [10,11].

In this context, this study investigated the effect of MetS on CRC incidence by conducting PS-based analyses considering age, alcohol consumption, smoking, high animal fat intake, obesity, a lack of dietary fiber intake, and a lack of physical activity, etc., which have been identified as modifiable risk factors for CRC [12,13,14,15,16,17,18,19], in a community-based prospective cohort in the Republic of Korea.

## 2. Materials and Methods

### 2.1. Data Source and Study Population

The Korean National Cancer Center Community (KNCCC) Cohort, as a community-based prospective cohort, was conducted by the KNCCC and included 16,304 men and women who resided in Changwon-si, Chuncheon-si, Chungju-si, Sancheong-gun, and Haman-gun in the Republic of Korea [20]. All participants were aged over 30 years, with the following average age at cohort entry: 58.6 ± 12.4 years for men (*N* = 6302) and 57.7 ± 13.3 for women (*N* = 10,002). The questionnaire survey was conducted by well-trained interviewers and included the following demographic information: age, sex, home region, education level, occupational history, marriage status, average household income, alcohol consumption, smoking status, physical activity, dietary intake, history of cancer, and exposure to pesticides. Additionally, the results of anthropometric measurements and clinical laboratory examinations were included. All study participants are followed through linkage to the Korean Central Cancer Registry for cancer incidence and mortality by 2016. This cohort was linked to mortality data of Statistics Korea for all participants by 2016. The study was approved by the KNCC Institutional Review Board (IRB No. NCC2016-0300).

A total of 6985 participants (2417 men and 4568 women) were eligible for the analysis after excluding 9319 participants who had a history of cancer before entry, did not participate in the nutrition survey, developed cancers other than CRC, and had missing data for MetS, alcohol consumption, smoking, physical activity, diet, and education (Figure 1). The study was approved by the KNCC Institutional Review Board (IRB No. NCC2016-0300). All participants had provided written informed consent.

### 2.2. Definition of CRC and MetS

The outcome of this study was CRC incidence; the type of cancer was coded as C18, C19, and C20 according to the International Classification of Diseases 10th edition (ICD-10).

In this study, a modified definition of MetS was applied. Since there is no information of waist circumference (WC) in this study, WC > 90 cm for men and >80 cm for women were substituted by body mass index (BMI) ≥ 25 kg/m^2^ referring to previous studies [21,22], and the other four components of MetS were applied from the National Cholesterol Education Program Adult Treatment Panel III (NCEP-ATP III) definition: hypertension (systolic blood pressure [SBP] ≥ 130 mmHg and diastolic blood pressure [DBP] ≥ 85 mmHg), low high-density lipoprotein (HDL) cholesterol level (<40 mg/dL for men and <50 mg/dL for women), high triglyceride (TG) level (≥150 mg/dL), and abnormal fasting blood sugar (FBS) level (≥110 mg/dL). A diagnosis of metabolic syndrome is rendered when more than three or above five conditions occur simultaneously [23].

### 2.3. Statistical Analyses

Descriptive analyses between subjects who developed CRC and those who did not were performed for continuous parameters using the Mann–Whitney U test and for categorical parameters using the Chi-square test and Fisher’s exact test.

Follow-up started at enrollment until a colorectal cancer diagnosis or censoring. Censoring occurred at date of death, or end of follow-up (31 December 2016).

To elucidate associations between MetS and risk of CRC and an association between each component of metabolic syndrome and CRC in both sexes, hazard ratios (HRs) and 95% confidence intervals (CIs) were estimated by the following statistical analyses using Cox hazard regression models: unadjusted and multiple regression, PS matching analysis, regression adjustment with the PS, and PS weighting methods. In these analyses, we included the following covariates known as the known risk factors of CRC [2,12,13,14,15,16,17,18,19,24,25]: age, alcohol consumption (non-drinkers, moderate drinkers [<24 g/day], heavy drinkers [≥24 g/day]), smoking status (non-smokers, moderate smokers [<20 pack-year], heavy smokers [≥20 pack-year]), moderate-intensity physical activity (days/week), frequency of fruit or vegetable intake (days/week), frequency of intake of beef or pork (days/month), education level (illiterate, middle school or less, high school, and college or more), and study area (Changwon-si, Chuncheon-si, Chungju-si, Sancheong-gun, and Haman-gun).

To perform PS-based methods, we needed first to calculate the PS. The PS is the probability that an individual would have MetS or abnormal TG levels based on personal demographic and lifestyle information, and it was obtained from the fit of a logistic regression model adjusted with all the covariates mentioned above.

To evaluate the balance in baseline characteristics in the dataset used for different regression models, we calculated the standardized mean differences, and values less than 0.1 were considered negligible differences. First, we applied a 1:1 case-control matching to the PS technique, which is the eighth digit to first digit greedy matching method [26]. This method may result in a drop-out of unmatched cases for the best matching. Detail on subjects in the matching analyses for MetS, abnormal TG levels, hypertension, obesity, abnormal high-density lipoprotein cholesterol levels, and abnormal fasting blood sugar levels was shown in the supplementary method. Second, we performed weighted Cox hazard regression models that were considered for an adjusted MetS effect after stratifying into 5 strata according to the quintiles of the estimated PS [27]. Additionally, a Cox hazard regression model that used the PS as a covariate (a continuous variable and a categorical variable by quintiles) was a simple PS method similar to traditional regression analysis. Lastly, there are two major PS weighting methods referred to as standardization methods that depend on the establishment of a standard population. One is the inverse probability-of-treatment weighted (IPTW) model [28]. It considers the overall study participants as the standard population, and it uses weights of (1/PS) for those with MetS and [1/(1 − PS)] for those without MetS. The other is the standardized mortality ratio weighted (SMRW) model. It regards those without MetS as the standard population and applies weights of 1 for those with MetS and [PS/(1 − PS)] for those without MetS.

Statistical analyses were carried out using SAS software, version 9.3 (SAS Institute, Cary, NC, USA) and R version 3.4.3 using the ‘tableone’ package [29] and were two-sided, with a significance level of *p* < 0.05.

## 3. Results

In total, 2417 men and 4568 women were included in this study (Figure 1). Table 1 shows the baseline characteristics of the study population. Among the study subjects, there were 57 men and 54 women newly diagnosed with CRC after the entry of this cohort and their median follow-up years were 4.76 (IQR, 2.91–7.79) for men and 5.55 (IQR, 3.06–7.53) for women, respectively. The mean age of the study subjects was approximately 60 years old (data not shown), which is older than the general Korean population. CRC cases were significantly older than non-cancer controls. In men, alcohol consumption was higher in CRC cases, i.e., there was a high percentage of non-drinkers in non-cancer controls. On the other hand, CRC cases for women took in more beef or pork per week than non-cancer controls. In this study, 524 men (21.7%) and 1297 women (28.4%) had at least three components of MetS at the entry of a cohort.

Appendix A show participants’ characteristics according to the presence of MetS and abnormal TG levels as well as the degree of imbalance among covariates. A statistically negligible difference in covariates was found in the following datasets: the matched, stratified, and IPTW datasets for MetS and abnormal TG levels in men; the matched and IPTW datasets for MetS in women; the matched, stratified and IPTW datasets for increased TG level in women.

Table 2 showed the associations of MetS and abnormal TG levels on CRC risk according to various analytical methods. In women, unadjusted and adjusted HRs between abnormal MetS and CRC risk were 2.33 (95% CI: 1.37, 3.97) and 2.12 (95% CI: 1.22, 3.68), respectively. The HR from the PS 1:1 matching analysis was 2.19 (95% CI: 1.10, 4.33) and that from the IPTW analysis was 2.03 (95% CI: 1.40, 2.95). Besides, unadjusted and adjusted HRs between abnormal TG levels and CRC were 2.27 (95% CI: 1.33, 3.87) and 2.06 (95% CI: 1.20, 3.55), respectively. The HRs from the PS 1:1 matched, stratified, and IPTW datasets were 2.08 (95% CI: 1.07, 4.02), 2.26 (95% CI: 1.32, 3.84), and 1.98 (95% CI: 1.36, 2.89), respectively. On the other hand, in men, all associations between MetS and abnormal TG levels and CRC risk were not significant.

Additionally, associations between four metabolic components except abnormal TG levels (i.e., hypertension, obesity, abnormal high-density lipoprotein cholesterol levels, and abnormal fasting blood sugar levels) and CRC risk are summarized in Appendix A. In women, all ORs between four metabolic components (hypertension, obesity, abnormal high-density lipoprotein cholesterol levels, and abnormal fasting blood sugar levels) and CRC risk were not consistently significant by various analyses. On the other hand, in men, only abnormal high-density lipoprotein cholesterol levels had a significant inverse association with CRC risk.

Besides, when associations between metabolic syndrome and the incidence of (a) colon cancer and (b) rectum cancer were evaluated, in women, there was an association between metabolic syndrome and rectum cancer (Table 3). The HRs from the PS 1:1 matched and IPTW datasets were 3.67 (95% CI: 1.03, 13.17) and 3.29 (95% CI: 1.29, 8.36), respectively, which were the only HRs estimated from datasets with balanced covariates (the degree of imbalance among covariates was not shown).

## 4. Discussion

To the best of our knowledge, our study is the first cohort study using PS-based methods to examine the effect of MetS on CRC incidence for both sexes in the Asian population. In this community-based cohort study in the Republic of Korea, we found an increased risk of CRC in women associated with MetS and abnormal TG level in both traditional Cox hazard regression and PS methods. In women, both MetS and abnormal TG level were associated with an approximately 2.0-fold to 2.5-fold increased risk of CRC. However, this study shows that there was no significant association between MetS and abnormal TG level and CRC risk in men no matter which analytic methods we performed.

As previously known, PS methods are reliable and provide excellent covariate balance, especially PS matching, although there are its cons which may lead to poor performance with few outcome events (stratification), drop-out of unmatched cases for the best matching (PS matching), and imprecise estimates of treatment effect (IPTW), etc. [30]. The meaningful findings of this study are consistent and have similar strengths of associations regardless of PS-based methods, although various methodologies were applied in statistical analysis. Thus, this study provides strong evidence on the relationship between MetS and abnormal TG level and CRC risk. However, it seems that previous research findings were inconsistent: a meta-analysis reported that there was a significant association between MetS and CRC in both sexes in cohort studies across populations, including the United States, European, Asian, and other populations (Relative risk (RR) for men, 1.25 [95% CI, 1.19, 1.32]; RR for women, 1.34, [95% CI, 1.09, 1.64]) [2]. In another meta-analysis, the results in cohort studies across populations showed that men with MetS had a significantly elevated risk of CRC, but women with MetS did not [4]. However, in the United States, a cohort study that examined for postmenopausal women showed a similar risk level to ours (HR, 2.15; 95% CI, 1.30, 3.53) [31]; the subsequent study recruited more participants but reported non-significant lower risks (HR, 1.16; 95% CI, 0.95–1.41) [32].

To discuss the association between MetS and CRC for the Asian population, two previous meta-analyses reported results using a couple of cohort studies for the Asian population: One showed that there was no association in cohort studies (RR for men, 1.10 [95% CI, 0.80, 1.51]; RR for women, 1.02, [95% CI, 0.76, 1.36]) [2]. Another also found that RRs for men and women were non-significant (RR for men, 1.23 [95% CI, 0.80, 1.88]; RR for women, 1.12, [95% CI, 0.86, 1.48]) [4]. In the case of the Republic of Korea, there were prior cohort studies about this association. Of these two studies using colon or colorectal adenoma risks as outcomes, one reported no association, but the other showed an association (adjusted HR, 1.28; 95% CI, 1.09–1.51) [6,7]. The subjects of these cohort studies underwent colonoscopy for a health examination in a large hospital and could, therefore, be generally regarded as individuals who are interested in the prevention of future disease and the pursuit of a healthy lifestyle. Two studies using the National Health Insurance Service–National Sample Cohort to represent the Korean population reported somewhat conflicting results: One found only a significantly increased risk of colon cancer, not rectum cancer, for men with MetS (HR, 1.40 [95% CI, 1.14, 1.71]), but did not find significant colon or rectum cancer risks for women [5]. The other found that MetS was associated with the development of CRC in both sexes (HR for men, 1.41 [95% CI, 1.37, 1.44]; HR for women, 1.23 [95% CI, 1.20, 1.27]) [33]. Our finding seems to be slightly different from previous studies above, as we have estimated using PS-based methods and the study subjects of our study were just community residents. Thus, we expect that this study might contribute to providing additional evidence that there is an association between MetS and an elevated risk of CRC in the Asian population, and future large cohort studies using PS-based methods could provide more definitive evidence.

In addition, our study shows that abnormal TG levels were associated with CRC risk in women. However, a previous review and meta-analysis reported that TG was not related to the risk of CRC [4]. Cohort studies in the United States and Japan also arrived at similar conclusions [31,34]. In the Republic of Korea, previous cohort studies showed the HRs of colorectal adenoma to be 1.19 (95% CI, 1.03, 1.37) and 0.76 (95% CI, 0.45, 1.27) [6,7] and the HR of CRC to be 1.15 (95% CI, 1.07, 1.23) [33]. Future studies are recommended as it remains controversial whether the risk of CRC or colorectal adenoma is elevated when individuals’ levels of TG are 150 mg/dL or higher.

As is well known, there are growing evidences that MetS is probably a risk factor for CRC, but the biological mechanism underlying this association remains to be clarified regardless of sex. Insulin resistance, systemic inflammation, oxidative stress, and higher leptin levels have been suggested as potential mechanisms that may explain the association between MetS and CRC [3]. Insulin increases cell proliferation and reduces apoptosis, which may lead to tumor development. Insulin also induces the overstimulation of receptors of insulin-like growth factor-1 and 2 (IGF-1 and IGF-2), a key promoter of tumor development. Besides, deteriorated metabolic status influences elevated levels of inflammatory cytokines such as interleukin 6 (IL-6), tumor necrosis factor alpha (TNF-α), C-reactive protein (CRP), and leptin hormone, which may be implicated in insulin resistance and tumor development. Further studies are required to elucidate the mechanism underlying the effect of each component of MetS on CRC.

The strengths of this study include its prospective nature (i.e., cohort design), the strong causality between MetS and CRC by using PS-based methods. The selection biases that are present in most observational studies may lead to a lack of causality in this study. To improve the causality between MetS and CRC incidence, we performed PS-based analyses in a community-based cohort. A major advantage of using PS-based methods in observational studies can minimize selection biases by balancing nonrandomized individuals’ data to reach the level of causality determined by randomized controlled trials. Recently, there have been some well-designed studies that have revealed associations between MetS and non-communicable diseases, including cancers, using PS-based analyses [35,36].

There were several limitations to this study. First, we could not observe the following confounders: the history of MetS or medications before cohort entry, individuals’ stressful events, menopause, the consumption of carbohydrate and starchy foods, etc. In addition, there is the possibility of information bias due to the use of a self-reported questionnaire. Second, we measured MetS only at the entry of this cohort study, so we could not estimate the risk of CRC due to changes in MetS over time. Third, we used BMI as a measure of the abdominal obesity of MetS due to the absence of WC data, although BMI and WC are slightly different in the pathological meaning of MetS. Forth, this study includes the bias due to the Asian population and there are limits of external validity. Although prospective cohort studies are greatly needed to determine causal associations between exposure and disease and may not require representativeness, the findings from this study need to be further investigated. Lastly, this study had limited statistical power due to the relatively small study sample. We could not evaluate the interaction between Mets and other risk factors (e.g., smoking, alcohol consumption, diet, and physical activities) on CRC risk. Furthermore, we found association between MetS and rectum cancer in women. However, due to the few rectum cancer cases, HRs by Cox regression methods and ORs by PS-based methods were significant but these CIs were quite wide. Thus, studies with larger sample sizes are needed to improve the statistical validation of the findings.

## 5. Conclusions

In conclusion, this study may provide additional evidence that deteriorated metabolic profiles increase the risk of CRC in women. We highlight the importance of effective population-level interventions for deteriorated metabolic profiles at the early stages.

## Figures and Tables

**Figure 1 ijerph-17-08687-f001:**
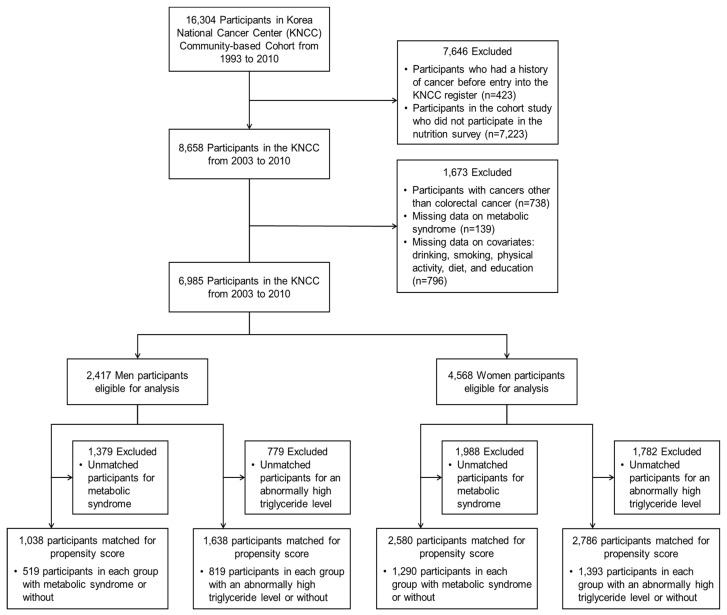
Flow diagram of the derivation of the study population.

**Table 1 ijerph-17-08687-t001:** Baseline characteristics of study participants.

Characteristics	Men			Women		
	Colorectal Cancer		*p*-Value	Colorectal Cancer		*p*-Value
	No (*N* = 2360)	Yes (*N* = 57)		No (*N* = 4514)	Yes (*N* = 54)	
Follow-up (years, Median (IQR))	10.43 (8.5–12.48)	4.76 (2.91–7.79)	<0.001	10.44 (9.48–12.88)	5.55 (3.06–7.53)	<0.001
Age (years, Mean ± SD)	59.66 ± 10.93	64.18 ± 8.67	0.002	59.95 ± 11.2	64.72 ± 9.19	0.001
Physical activity (days/week, Mean ± SD)	3.97 ± 2.86	4.26 ± 2.7	0.754	2.99 ± 2.96	3.2 ± 3.02	0.60
Intake of fruits or vegetables (days/week, Mean ± SD)	5.43 ± 1.44	5.19 ± 1.57	0.28	5.51 ± 1.46	5.04 ± 1.8	0.073
Intake of beef or pork (days/week, Mean ± SD)	2.55 ± 1.4	2.44 ± 1.49	0.732	1.75 ± 1.33	1.31 ± 1.33	0.012
Alcohol consumption [*N*(%)]						
Non-drinkers	640 (27.12)	7 (12.28)	0.03	3598 (79.71)	45 (83.33)	0.796
Moderate drinkers (<24 g/day)	797 (33.77)	26 (45.61)		797 (17.66)	8 (14.81)	
Heavy drinkers (≥24 g/day)	923 (39.11)	24 (42.11)		119 (2.64)	1 (1.85)	
Smoking status [*N*(%)]						
Non-smokers	483 (20.47)	12 (21.05)	0.952	4186 (92.73)	52 (96.3)	0.894
Moderate smokers (<20 pack-year)	669 (28.35)	17 (29.82)		263 (5.83)	2 (3.7)	
Heavy smokers (≥20 pack-year)	1208 (51.19)	28 (49.12)		65 (1.44)	0 (0)	
Education level [*N*(%)]						
Illiterate	222 (9.41)	6 (10.53)	0.448	1445 (32.01)	24 (44.44)	0.24
Middle school or less	1443 (61.14)	39 (68.42)		2505 (55.49)	26 (48.15)	
High school	484 (20.51)	10 (17.54)		438 (9.7)	3 (5.56)	
College or more	211 (8.94)	2 (3.51)		126 (2.79)	1 (1.85)	
Residential area [*N*(%)]						
Sancheong-gun	1270 (53.81)	23 (40.35)	0.209	2380 (52.72)	21 (38.89)	0.072
Changwon-si	485 (20.55)	16 (28.07)		867 (19.21)	13 (24.07)	
Chooncheon-si	167 (7.08)	7 (12.28)		438 (9.7)	3 (5.56)	
Choongjoo-si	281 (11.91)	8 (14.04)		558 (12.36)	11 (20.37)	
Haman-gun	157 (6.65)	3 (5.26)		271 (6)	6 (11.11)	
Metabolic syndrome [*N*(%)]						
No (No. of components of MetS < 3)	1848 (78.31)	45 (78.95)	0.907	3243 (71.84)	28 (51.85)	0.001
Yes (No. of components of MetS ≥ 3)	512 (21.69)	12 (21.05)		1271 (28.16)	26 (48.15)	
Blood pressure [*N*(%)]						
Normal BP	996 (42.2)	21 (36.84)	0.418	2017 (44.68)	19 (35.19)	0.163
High BP	1364 (57.8)	36 (63.16)		2497 (55.32)	35 (64.81)	
BMI [*N*(%)]						
<25 kg/m2	1667 (70.64)	40 (70.18)	0.94	2840 (62.92)	33 (61.11)	0.785
≥25 kg/m2	693 (29.36)	17 (29.82)		1674 (37.08)	21 (38.89)	
HDL cholesterol [*N*(%)]						
Normal HDL	1913 (81.06)	54 (94.74)	0.009	2309 (51.15)	26 (48.15)	0.661
Low HDL	447 (18.94)	3 (5.26)		2205 (48.85)	28 (51.85)	
Triglyceride level [*N*(%)]						
Normal TG	1552 (65.76)	40 (70.18)	0.488	3148 (69.74)	27 (50)	0.002
High TG	808 (34.24)	17 (29.82)		1366 (30.26)	27 (50)	
FBS [*N*(%)]						
Normal FBS	2047 (86.74)	46 (80.7)	0.186	4042 (89.54)	49 (90.74)	0.775
High FBS	313 (13.26)	11 (19.3)		472 (10.46)	5 (9.26)	

IQR, interquartile range; SD, standard deviation; SBP, systolic blood pressure; DBP, diastolic blood pressure; BMI, body mass index; HDL, high-density lipoproteins; FBS, fasting blood sugar. High BP: SBP ≥ 130 mmHg or DBP ≥ 85 mmHg, Low HDL: <40 mg/dL for men, <50 mg/dL for women, High TG: ≥150 mg/dL ≥110 mg/dL, High FBS: ≥110 mg/dL.

**Table 2 ijerph-17-08687-t002:** Associations between (a) metabolic syndrome, (b) triglyceride level and colorectal cancer risk.

	Total				Men				Women			
Methods	Cases (*N*)	Controls (*N*)	HR (OR) (95% CI)	*p*-Value	Cases (*N*)	Controls (*N*)	HR (OR) (95% CI)	*p*-Value	Cases (*N*)	Controls (*N*)	HR (OR) (95% CI)	*p*-Value
(a) Metabolic syndrome												
Cox hazard regression												
Unadjusted	111	6874	1.46 (0.99, 2.16)	0.060	57	2360	0.94 (0.50, 1.78)	0.856	54	4514	2.33 (1.37, 3.97)	0.002
Multivariable ^(a)^	111	6874	1.55 (1.04, 2.33)	0.033	57	2360	1.04 (0.54, 1.99)	0.908	54	4514	2.12 (1.22, 3.68)	0.008
PS-based logistic regression												
Matched for PS	67	3573	1.32 (0.81, 2.13)	0.266	25	1013	0.93 (0.42, 2.03)	0.847	38	2542	2.19 (1.10, 4.33)	0.025
Stratification into 5 strata by PS	111	6862	1.48 (1.00, 2.19)	0.050	57	2345	1.06 (0.56, 2.02)	0.859	53	4505	2.23 (1.32, 3.77)	0.003
Regression adjusted with PS												
as a continuous term	111	6874	1.45 (0.97, 2.16)	0.071	57	2360	1.05 (0.55, 2.01)	0.878	54	4514	2.03 (1.17, 3.53)	0.012
as a quintile term	111	6874	1.48 (0.99, 2.22)	0.054	57	2360	1.02 (0.54, 1.95)	0.947	54	4514	2.07 (1.20, 3.58)	0.009
Weighted models												
IPTW model	111	6874	1.43 (1.11, 1.85)	0.007	57	2360	1.06 (0.73, 1.52)	0.772	54	4514	2.03 (1.40, 2.95)	<0.001
SMRW model	111	6874	1.44 (1.04, 2.01)	0.031	57	2360	0.92 (0.52, 1.64)	0.780	54	4514	2.48 (1.63, 3.75)	<0.001
(b) Triglyceride level												
Cox hazard regression												
Unadjusted	111	6874	1.39 (0.95, 2.03)	0.090	57	2360	0.79 (0.45, 1.39)	0.416	54	4514	2.27 (1.33, 3.87)	0.003
Multivariable ^(a)^	111	6874	1.33 (0.91, 1.95)	0.145	57	2360	0.84 (0.47, 1.5)	0.557	54	4514	2.06 (1.2, 3.55)	0.009
PS-based logistic regression												
Matched for PS	76	4360	1.36 (0.86, 2.14)	0.191	32	1606	1.11 (0.55, 2.21)	0.777	40	2746	2.08 (1.07, 4.02)	0.031
Stratification into 5 strata by PS	111	6866	1.3 (0.89, 1.9)	0.168	57	2354	0.82 (0.47, 1.45)	0.498	53	4501	2.26 (1.32, 3.84)	0.003
Regression adjusted with PS												
as a continuous term	111	6874	1.28 (0.87, 1.88)	0.210	57	2360	0.85 (0.48, 1.51)	0.570	54	4514	2.02 (1.18, 3.48)	0.011
as a quintile term	111	6874	1.28 (0.87, 1.88)	0.202	57	2360	0.82 (0.46, 1.46)	0.504	54	4514	2.03 (1.18, 3.50	0.010
Weighted models												
IPTW model	111	6874	1.28 (0.99, 1.66)	0.063	57	2360	0.85 (0.58, 1.23)	0.380	54	4514	1.98 (1.36, 2.89)	<0.001
SMRW model	111	6874	1.44 (1.05, 1.97)	0.025	57	2360	0.79 (0.48, 1.30	0.350	54	4514	2.42 (1.6, 3.66)	<0.001

HR, hazard ratio; CI, confidence interval; PS, propensity score; IPTW, inverse probability-of-treatment weighted; SMRW, standardized mortality ratio weighted. ^(a)^ Adjusted by age, sex (in case of total) education, smoking status, alcohol consumption, physical activity, frequency of intake of fruits or vegetables, frequency of intake of red meats, and residential area.

**Table 3 ijerph-17-08687-t003:** Associations between metabolic syndrome and the incidence of (a) colon cancer and (b) rectum cancer in women.

	*N* (Cases/Controls)	HR(OR)	95% CI	*p* Value
(a) Colon cancer (C18–C19)				
Cox hazard regression				
Unadjusted	4549 (35/4514)	1.88	0.96, 3.68	0.064
Multivariable ^(a)^	4549 (35/4514)	1.71	0.86, 3.43	0.128
PS-based logistic regression				
Matched for PS	2564 (25/2539)	1.52	0.69, 3.39	0.302
Stratification into 5 strata by PS	4539 (34/4505)	1.81	0.93, 3.5	0.079
Regression adjusted with PS				
as a continuous term	4549 (35/4514)	1.6	0.8, 3.19	0.184
as a quintile term	4549 (35/4514)	1.63	0.82, 3.24	0.164
Weighted models				
IPTW model	4549 (35/4514)	3.29	1.29, 8.36	0.012
SMRW model	4549 (35/4514)	1.92	1.12, 3.28	0.018
(b) Rectum cancer (C20)				
Cox hazard regression				
Unadjusted	4533 (19/4514)	3.47	1.4, 8.64	0.007
Multivariable ^(a)^	4533 (19/4514)	3.25	1.28, 8.25	0.013
PS-based logistic regression				
Matched for PS	2554 (14/2540)	3.67	1.03, 13.17	0.046
Stratification into 5 strata by PS	4522 (19/4503)	3.16	1.3, 7.64	0.011
Regression adjusted with PS				
as a continuous term	4533 (19/4514)	3.19	1.25, 8.16	0.015
as a quintile term	4533 (19/4514)	3.29	1.29, 8.36	0.012
Weighted models				
IPTW model	4533 (19/4514)	3.29	1.29, 8.36	0.012
SMRW model	4533 (19/4514)	4.13	2.12, 8.06	<0.001

HR, hazard ratio; CI, confidence interval; PS, propensity score; IPTW, inverse probability-of-treatment weighted; SMRW, standardized mortality ratio weighted. ^(a)^ Adjusted by age, education, smoking status, alcohol consumption, physical activity, frequency of intake of fruits or vegetables, frequency of intake of red meats, and residential area.

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
