# Peer review of "Metabolic Syndrome and Colorectal Cancer Risk: Results of Propensity Score-Based Analyses in a Community-Based Cohort Study"

_ijerph, 2020, doi:10.3390/ijerph17228687_

Round 1

Reviewer 1 Report

The authors conducted the cohort study to elucidate the association between MetS and CRC. PS based analysis was used to calculate less-biased estimates. The association of MetS and CRC have still been inconsistent, therefore, the study might be contributed to clarify the problem. However, more discussions concerning causal inferences should be necessary.

1. CRC and MetS share some common risk factors (i.g. smoking, alcohol, obesity, diabetes, excercise). I think the risk factors cause MetS, then MetS affects CRC. At the same time, the risk factors might directly cause CRC. Thus, two causal pathways (risk factors-MetS-CRC and risk factors-CRC) should be considered when estimating the association between MetS and CRC.

2. When the two pathways were considered, effects of the risk factors might be different between subjects with and without MetS. Was there interaction between them?

3. MetS was associated with CRC only among females. Is there any mechanisms or evidences supporting the sex difference?

4. Eligibility criteria was different between main text and figure 1. Male subjects were excluded from the study in figure 1, but they were not excluded in main text.

5. Were those who took medications for MetS included in the study? If they were included, they might be classified as non-MetS group, because their examination levels were controlled.

6. BMI was used as a surrogate of waist circumference. Threshold of the BMI should be defined in methods section.

7. I guessed the definition of subjects with MetS was having 3 or more conditions from high BMI, HT, low HDL, high TG and high FBS, however, the definition was not clearly specified. In addition, TG was assessed separately. Was TG used to define MetS?

8. The association between CRC and TG were assessed separately with MetS. High TG was one of the components of MetS. The reason why TG should be assessed separately should be explained.

9. Were PS for MetS and TG level calculated using Cox regression or logistic regression? Although the authors described as "the PS was obtained from the fit of a Cox regression model (page4, line 111)", MetS and TG seemed not to have time variables.

Reviewer 2 Report

Comments

  1. In this study, authors attempt to draw a relationship between MetS, CRC using PS methods especially comparing among men and women. The study is based on KNCC. This study is relevant given the preventative nature of CRC, identifying the potentially modifiable risk factors for CRC especially from the public health point of view. 
  2. Abstract-Results can be simplified further as there are a lot of numbers that are making it difficult for the reader to grasp the findings of the study. 
  3. Introduction: Authors appropriately mentioned the importance of CRC. It is highly recommended to mention the barriers of the CRC screening receipt especially modifiable ones (among the insured and non-insured population) (see PMID: 30191078, PMID: 31455888). And connecting these with potential public health measures to reduce the cancer burden 
  4. In the flow diagram→ it seems like only 4,665 women are included (are men excluded)-need further clarification on this.
  5. Methods: It is suggested to convert some of the information into the supplementary section. Highly recommended for the journal to have a statistician review of the results and methods
  6. Results: Authors state: 2776 men and 4,844 are noted in the study in figure 1-which is not clear. Suggested to bold or highlight these numbers. Table 1: Needs to be cut down (especially some of the residential area, which can be in the supplementary section) Further simplification→ making it as Normal BP/high, Normal TG/ high TG can be performed. Note: authors noted BMI of >.7kg/m2 which is unclear.
  7. Tables 2 and 3 are very busy and difficult to read. Either reduce the content and move to the supplementary section
  8. Discussion: Pros and cons of the PS can be mentioned at the end of the manuscript. 
  9. Limitations: Please include the bias due to the Asian population and limits of its external validity.  However, given the higher incidence of Mets in the Western population, these findings could be even more significant in obese individuals but this needs to be further studied.

Round 2

Reviewer 1 Report

The authors responded to all comments from reviewers. The manuscript was well revised. I have no additional comments.